# Super-Convergence: Very Fast Training of Residual Networks Using Large Learning Rates

## Abstract

In this paper, we show a phenomenon, which we named "super-convergence", where residual networks can be trained using an order of magnitude fewer iterations than is used with standard training methods. The existence of super-convergence is relevant to understanding why deep networks generalize well. One of the key elements of super-convergence is training with cyclical learning rates and a large maximum learning rate. Furthermore, we present evidence that training with large learning rates improves performance by regularizing the network. In addition, we show that super-convergence provides a greater boost in performance relative to standard training when the amount of labeled training data is limited. We also derive a simplification of the Hessian Free optimization method to compute an estimate of the optimal learning rate. The architectures to replicate this work will be made available upon publication.

## 1 Introduction

While deep neural networks have achieved amazing successes in a range of applications, understanding why stochastic gradient descent (SGD) works so well remains an open and active area of research. This paper provides unique empirical evidence that supports the theories in some papers but not others. Specifically, we show that, for certain datasets, residual network architectures (He et al., 2016), and hyper-parameter values, using very large learning rates with the cyclical learning rate (CLR) method (Smith, 2015; 2017) can speed up training by an order of magnitude. Analogous to the phenomenon of super-conductivity that only happens in limited circumstances and provides theoretical insights of materials, we named this phenomenon "super-convergence." While super-convergence might be of some practical value, the primary purpose of this paper is to provide empirical support and theoretical insights to the active discussions in the literature on SGD and understanding generalization.

Figure 1a provides a comparison of test accuracies from a super-convergence example and the result of a typical (piecewise constant) training regime for Cifar-10, both using a 56 layer residual network architecture. Piecewise constant training reaches a peak accuracy of 91.2% after approximately 80,000 iterations, while the super-convergence method reaches a higher accuracy (92.4%) after only 10,000 iterations. Figure 1b shows the results for a range of CLR stepsize values, where training lasted only one cycle. This modified learning rate schedule achieves a higher final test accuracy (92.1%) than typical training (91.2%) after only 6,000 iterations. In addition, as the total number of iterations increases from 2,000 to 20,000, the final accuracy improves from 89.7% to 92.7%.

The contributions of this paper are:

1. We demonstrate a new training phenomenon and systematically investigate it.

2. We show evidence that large learning rates regularize the trained network and hypothesize that this allows SGD to find a flat local minima that generalizes well.

3. We derive a simplification of the second order, Hessian-free optimization method to estimate optimal learning rates which demonstrates that large learning rates find wide, flat minima.

4. We demonstrate that the effects of super-convergence are increasingly dramatic when less labeled training data is available.

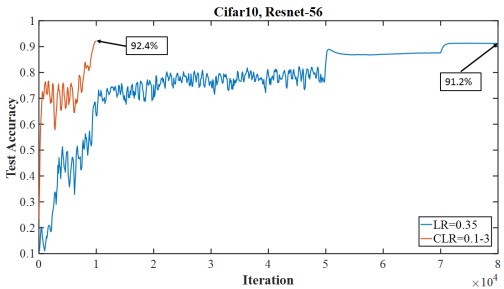
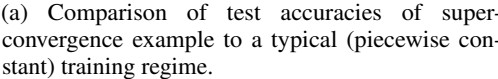

(a) Comparison of test accuracies of super-convergence example to a typical (piecewise constant) training regime.

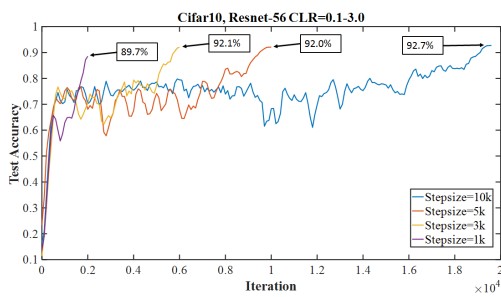

(b) Comparison of test accuracies of super-convergence for a range of stepsizes.

Figure 1: Examples of super-convergence with Resnet-56 on Cifar-10.

## 2 BACKGROUND

In this paper, when we refer to a typical, standard, or a piecewise-constant training regime, it means the practice of using a global learning rate, ($\approx 0.1$), for many epochs, until the test accuracy plateaus, and then continuing to train with a lower learning rate decreased by a factor of $0.1$. This process of reducing the learning rate and continuing to train is often repeated two or three times.

There exists extensive literature on stochastic gradient descent (SGD) (see Goodfellow et al. (2016) and Bottou (2012)) which is relevant to this work. Also, there exists a significant amount of literature on the loss function topology of deep networks (see Chaudhari et al. (2016) for a review of the literature). This paper contains a discussion of the loss function topology which follows from the work of Goodfellow et al. (2014) on characterizing the landscape. Our use of large learning rate values is in contrast to suggestions in the literature of a maximum learning rate value Bottou et al. (2016). Methods for adaptive learning rates have also been an active area of research. This paper uses a simplification of the second order Hessian-Free optimization (Martens, 2010) to estimate optimal values for the learning rate. In addition, we utilize some of the techniques described in Schaul et al. (2013) and Gulcehre et al. (2017). Also, we show that adaptive learning rate methods such as Nesterov momentum (Sutskever et al., 2013; Nesterov, 1983), AdaDelta (Duchi et al., 2011), AdaGrad (Zeiler, 2012), and Adam (Kingma & Ba, 2014) do not use sufficiently large learning rates when they are effective nor do they lead to super-convergence without using CLR. A warmup learning rate strategy (He et al., 2016; Goyal et al., 2017) could be considered a discretization of CLR, which was also recently suggested in (Jastrzębski et al., 2017). We note that Loshchilov & Hutter (2016) subsequently proposed a similar method to CLR, which they call SGDR. The SGDR method uses a sawtooth pattern with a cosine followed by a jump back up to the original value. Experiments show that it is not possible to observe the super-convergence phenomenon when using their pattern.

Our work is intertwined with several active lines of research in the deep learning research community, including a lively discussion on stochastic gradient descent (SGD) and understanding why solutions generalize so well, research on SGD and the importance of noise for generalization, and the generalization gap between small and large mini-batches. We defer our discussion of these lines of research to Section 7 where we can compare to our empirical results and theoretical insights.

## 3 SUPER-CONVERGENCE

In this work, we use cyclical learning rates (CLR) and the learning rate range test (LR range test) which were first suggested by Smith (2015) and later updated in Smith (2017). To use CLR, one specifies minimum and maximum learning rate boundaries and a stepsize. The stepsize is the number of iterations used for each step and a cycle consists of two such steps – one in which the learning rate increases and the other in which it decreases. Smith (2015) tested numerous ways to vary the learning rate between the two boundary values, found them to be equivalent and therefore recommended the simplest, which is letting the learning rate change linearly (similarly Jastrzębski et al. (2017) suggest discrete jumps). Please note that cyclical noise (via learning rates, batch size, etc.) is obtained by **combining curriculum learning (Bengio et al., 2009) and simulated annealing (Aarts & Korst, 1988)**, both of which have a long history of use in deep learning.

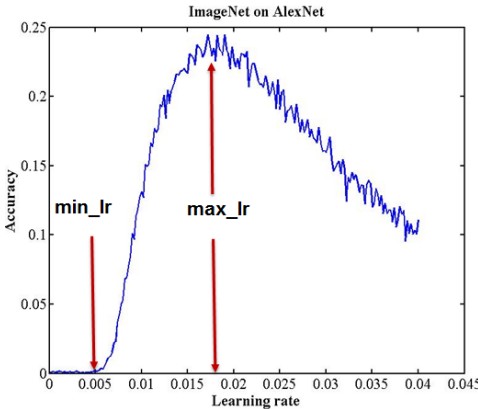

(a) Typical learning rate range test result where there is a peak to indicate max_lr.

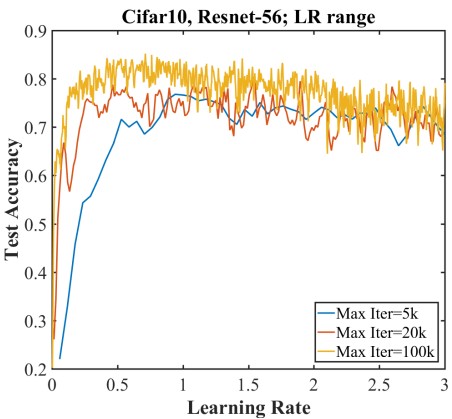

(b) Learning rate range test result with the Resnet-56 architecture on Cifar-10.

Figure 2: Comparison of learning rate range test results.

The LR range test can be used to determine if super-convergence is possible for an architecture. In the LR range test, training starts with a zero or very small learning rate which is slowly increased linearly throughout a pre-training run. This provides information on how well the network can be trained over a range of learning rates. Figure 2a shows a typical curve from a LR range test, where the test accuracy has a distinct peak.[1] When starting with a small learning rate, the network begins to converge and, as the learning rate increases, it eventually becomes too large and causes the training/test accuracy to decrease. The learning rate at this peak is the largest value to use as the maximum learning rate bound when using CLR. The minimum learning rate can be chosen by dividing the maximum by a factor of 3 or 4. The optimal initial learning rate for a typical (piecewise constant) training regime usually falls between these minimum and maximum values.

If one runs the LR range test for Cifar-10 on a 56 layer residual networks, one obtains the curves shown in Figure 2b. Please note that learning rate values up to 3.0 were tested, which is at least an order of magnitude lager than typical values of the learning rate. The test accuracy remains consistently high over this unusually long range of large learning rates. This unusual behavior motivated our experimentation with much higher learning rates, and we believe that such behavior during a LR range test is indicative of potential for super-convergence. The three curves in this figure are for runs with a maximum number of iterations of 5,000, 20,000, and 100,000, where the test accuracy remains consistently high for a long range of large learning rate values.

Figure 3a provides an example of transversing the loss function topology.[2] This figure helps give an intuitive understanding of how super-convergence happens. The blue line in the Figure represents the trajectory of the training while converging and the x's indicate the location of the solution at each iteration and indicates the progress made during the training. In early iterations, the learning rate must be small in order for the training to make progress in an appropriate direction. The Figure also shows that significant progress is made in those early iterations. However, as the slope decreases so does the amount of progress per iteration and little improvement occurs over the bulk of the iterations. Figure 3b shows a close up of the final parts of the training where the solution maneuvers through a valley to the local minimum within a trough.

Cyclical learning rates are well suited for training when the loss topology takes this form. The learning rate initially starts small to allow convergence to begin. As the network traverses the flat valley, the learning rate is large, allowing for faster progress through the valley. In the final stages of the training, when the training needs to settle into the local minimum (as seen in Figure 3b), the learning rate is once again reduced to its original small value.

---

[1]Figure reproduced from Smith (2017) with permission.
[2]Figure reproduced from Goodfellow et al. (2014) with permission.

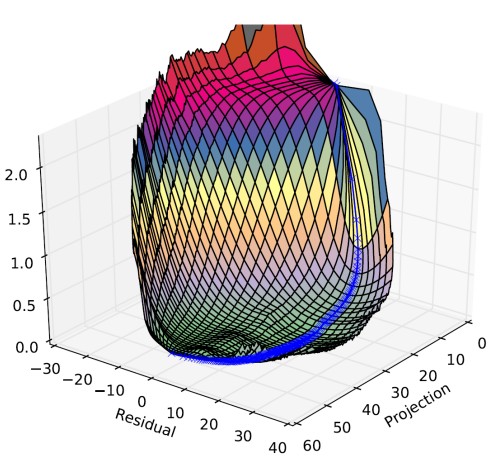

(a) Visualization of how training transverses a loss function topology.

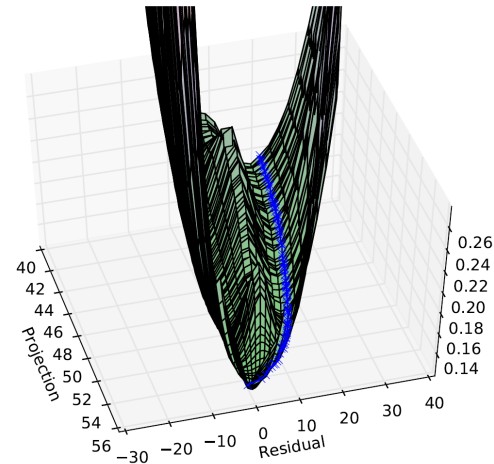

(b) A close up of the end of the training for the example in Figure 3a.

Figure 3: The 3-D visualizations from Goodfellow et al. (2014). The z axis represents the loss potential.

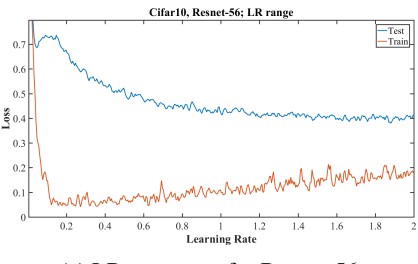

(a) LR range test for Resnet-56.

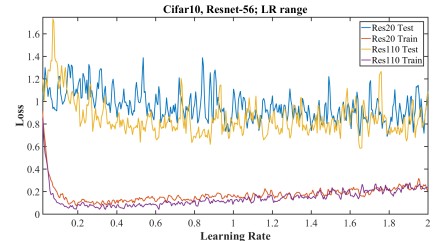

(b) LR range tests for Resnet-20 and Resnet-110.

Figure 4: Evidence of regularization with large learning rates: decreasing generalization error as the learning rate increases from 0.3 to 1.5.

## 4 LARGE LEARNING RATE REGULARIZATION

In our experiments, we only observed super-convergence with large learning rates. The LR range test reveals evidence of regularization through results shown in Figure 4a. Figure 4a shows an increasing training loss and decreasing test loss while the learning rate increases from approximately 0.2 to 2.0 when training with the Cifar-10 dataset and a Resnet-56 architecture, which implies that regularization is occurring while training with these large learning rates. Similarly, Figure 4b presents the training and test loss curves for Resnet-20 and Resnet-110, where one can see the same decreasing generalization error. In addition, we ran the LR range test on residual networks with $l$ layers, where $l = 20 + 9n$; for $n = 0, 1, ...10$ and obtained similar results.

There is additional evidence that large learning rates are regularizing the training. As shown in Figure 1a, the final test accuracy results from a super-convergence training is demonstrably better than the accuracy results from a typical training method. In the literature, this type of improvement in the final training accuracy is often taken as evidence of regularization. Furthermore, others show that large learning rates leads to larger gradient noise, which leads to better generalization (i.e., Jastrzębski et al. (2017); Smith et al. (2017)).

## 5 ESTIMATING OPTIMAL LEARNING RATES

Gradient or steepest descent is an optimization method that uses the slope as computed by the derivative to move in the direction of greatest negative gradient to iteratively update a variable. That

is, given an initial point $x_0$, gradient descent proposes the next point to be:

$$x = x_0 - \epsilon \bigtriangledown_x f(x) \tag{1}$$

where $\epsilon$ is the step size or learning rate . If we denote the parameters in a neural network (i.e., weights) as $\theta \in R^N$ and $f(\theta)$ is the loss function, we can apply gradient descent to learn the weights of a network; i.e., with input $x$, a solution $y$, and non-linearity $\sigma$:

$$y = f(\theta) = \sigma(W_l \sigma(W_{l-1} \sigma(W_{l-2}...\sigma(W_0 x + b_0)... + b_l) \tag{2}$$

where $W_l \in \theta$ are the weights for layer $l$ and $b_l \in \theta$ are biases for layer $l$.

The Hessian-free optimization method (Martens, 2010) suggests a second order solution that utilizes the slope information contained in the second derivative (i.e., the derivative of the gradient $\bigtriangledown_\theta f(\theta)$). From Martens (2010), the main idea of the second order Newton's method is that the loss function can be locally approximated by the quadratic as:

$$f(\theta) \approx f(\theta_0) + (\theta - \theta_0)^T \bigtriangledown_\theta f(\theta_0) + \frac{1}{2}(\theta - \theta_0)^T H (\theta - \theta_0) \tag{3}$$

where $H$ is the Hessian, or the second derivative matrix of $f(\theta_0)$. Writing Equation 1 to update the parameters at iteration $i$ as:

$$\theta_{i+1} = \theta_i - \epsilon \bigtriangledown_\theta f(\theta_i) \tag{4}$$

allows Equation 3 to be re-written as:

$$f(\theta_i - \epsilon \bigtriangledown_\theta f(\theta_i)) \approx f(\theta_i) + (\theta_{i+1} - \theta_i)^T \bigtriangledown_\theta f(\theta_i) + \frac{1}{2}(\theta_{i+1} - \theta_i)^T H (\theta_{i+1} - \theta_i) \tag{5}$$

In general it is not feasible to compute the Hessian matrix, which has $\Omega(N^2)$ elements, where $N$ is the number of parameters in the network, but it is unnecessary to compute the full Hessian. The Hessian expresses the curvature in all directions in a high dimensional space, but the only relevant curvature direction is in the direction of steepest descent that SGD will traverse. This concept is contained within Hessian-free optimization, as Martens (2010) suggests a finite difference approach for obtaining an estimate of the Hessian from two gradients:

$$H(\theta) = \lim_{\delta \to 0} \frac{\bigtriangledown f(\theta + \delta) - \bigtriangledown f(\theta)}{\delta} \tag{6}$$

where $\delta$ should be in the direction of the steepest descent. The AdaSecant method (Gulcehre et al., 2014; 2017) builds an adaptive learning rate method based on this finite difference approximation as:

$$\epsilon^* \approx \frac{\theta_{i+1} - \theta_i}{\bigtriangledown f(\theta_{i+1}) - \bigtriangledown f(\theta_i)} \tag{7}$$

where $\epsilon^*$ represents the optimal learning rate for each of the neurons. Utilizing Equation 4, we rewrite Equation 7 in terms of the differences between the weights from three sequential iterations as:

$$\epsilon^* = \epsilon \ \frac{\theta_{i+1} - \theta_i}{2\theta_{i+1} - \theta_i - \theta_{i+2}} \tag{8}$$

where $\epsilon$ on the right hand side is the learning rate value actually used in the calculations to update the weights. Equation 8 is an expression for an adaptive learning rate for each weight update. We borrow the method in Schaul et al. (2013) to obtain an estimate of the global learning rate from the weight specific rates by summing over the numerator and denominator, with one minor difference. In Schaul et al. (2013) their expression is squared, leading to positive values – therefore we sum the absolute values of each quantity to maintain positivity (using the square root of the sum of squares of the numerator and denominator of Equation 8 leads to similar results).

For illustrative purposes, we computed the optimal learning rates from the weights of every iteration using Equation 8 for two runs: first when the learning rate was a constant value of $0.1$ and second with CLR in the range of $0.1 - 3$ with a stepsize of 5,000 iterations. Since the computed learning rate exhibited rapid variations, we computed a moving average of the estimated learning rate as $LR = \alpha \epsilon^* + (1 - \alpha) LR$ with $\alpha = 0.1$ and the results are shown in Figure 5a for the first 300 iterations. This curve qualitatively shows that the optimal learning rates should be in the range of 2 to 4 for this architecture. In Figure 5b, we used the weights as computed every 10 iterations and ran the

| # training samples | Policy (Range) | BN MAF | Total Iterations | Accuracy (%) |
|---|---|---|---|---|
| 40,000 | PC-LR=0.35 | 0.999 | 80,000 | 89.1 |
| 40,000 | CLR (0.1-3) | 0.95 | 10,000 | 91.1 |
| 30,000 | PC-LR=0.35 | 0.999 | 80,000 | 85.7 |
| 30,000 | CLR (0.1-3) | 0.95 | 10,000 | 89.6 |
| 20,000 | PC-LR=0.35 | 0.999 | 80,000 | 82.7 |
| 20,000 | CLR (0.1-3) | 0.95 | 10,000 | 87.9 |
| 10,000 | PC-LR=0.35 | 0.999 | 80,000 | 71.4 |
| 10,000 | CLR (0.1-3) | 0.95 | 10,000 | 80.6 |
| 50,000 | CLR (0.1-3.5) | 0.95 | 10,000 | 92.1 |
| 50,000 | CLR (0.1-3) | 0.95 | 10,000 | 92.4 |
| 50,000 | CLR (0.1-2.5) | 0.95 | 10,000 | 92.3 |
| 50,000 | CLR (0.1-2) | 0.95 | 10,000 | 91.7 |
| 50,000 | CLR (0.1-1.5) | 0.95 | 10,000 | 90.9 |
| 50,000 | CLR (0.1-1) | 0.95 | 10,000 | 91.3 |
| 50,000 | CLR (0.1-3) | 0.97 | 20,000 | 92.7 |
| 50,000 | CLR (0.1-3) | 0.95 | 10,000 | 92.4 |
| 50,000 | CLR (0.1-3) | 0.93 | 8,000 | 91.7 |
| 50,000 | CLR (0.1-3) | 0.90 | 6,000 | 92.1 |
| 50,000 | CLR (0.1-3) | 0.85 | 4,000 | 91.1 |
| 50,000 | CLR (0.1-3) | 0.80 | 2,000 | 89.7 |

Table 1: Comparison of final accuracy results for various training regimes of Resnet-56 on Cifar-10. BN MAF is the value use for the $moving\_average\_fraction$ parameter with batch normalization. PC-LR is a standard piecewise constant learning rate policy described in Section 2 with an initial learning rate of 0.35.

learning rate estimation for 10,000 iterations. An interesting divergence happens here: when keeping the learning rate constant, the learning rate estimate initially spikes to a value of about 3 but then drops down near $0.2$. On the other hand, the learning rate estimate using CLR remains high until the end where it settles down to a value of about $0.5$. The large learning rates indicated by these Figures is caused by small values of our Hessian approximation and small values of the Hessian implies that SGD is finding flat and wide local minima.

In this paper we do not perform a full evaluation of the effectiveness of this technique as it is tangential to the theme of this work. We only use this method here to demonstrate that training with large learning rates are indicated by this approximation. We leave a full assessment and tests of this method to estimate optimal adaptive learning rates as future work.

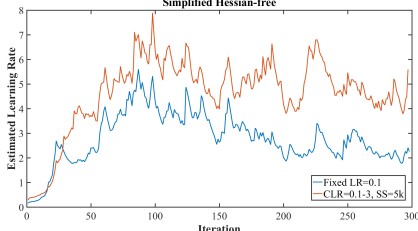
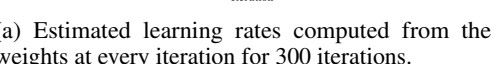
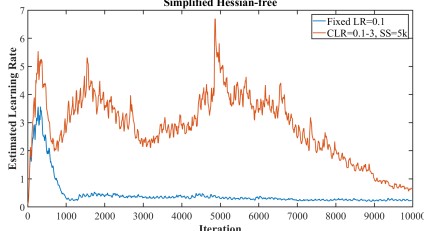

(a) Estimated learning rates computed from the weights at every iteration for 300 iterations.

(b) Estimated learning rates computed from the weights at every 10 iterations for 10,000 iterations.

Figure 5: Estimated learning rate from the simplified Hessian-free optimization (see text for additional information).

## 6   EXPERIMENTS AND ANALYSIS

This section highlights a few of our more significant experiments. Additional experiments and details of our architecture are illustrated in the Supplemental Section.

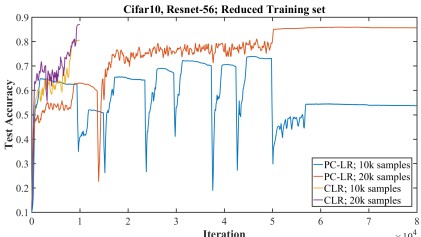

(a) Comparison of test accuracies for Cifar-10 with limited training samples.

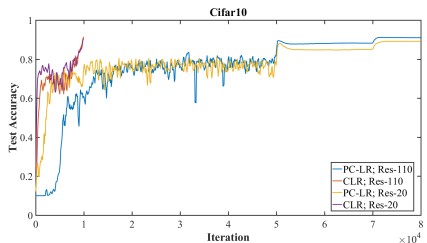

(b) Comparison of test accuracies for Resnet-20 and Resnet-110.

Figure 6: Comparisons of super-convergence to typical training outcome with piecewise constant learning rate schedule.

Figure 6a provides a comparison of super-convergence with a reduced number of training samples. When the amount of training data is limited, the gap in performance between the result of standard training and super-convergence increases. With a piecewise constant learning rate schedule the training encounters difficulties and diverges along the way. On the other hand, a network trained with specific CLR parameters exhibits super-convergence and trains without difficulties. The highest accuracies attained using standard learning rate schedules are listed in Table 1 and super-convergence test accuracy is 1.2%, 5.2%, and 9.2% better for 50,000, 20,000, and 10,000 training cases, respectively. Hence, super-convergence becomes more beneficial when training data is more limited.

We also ran experiments with Resnets with a number of layers in the range of 20 to 110 layers; that is, we ran experiments on residual networks with $l$ layers, where $l = 20 + 9n$; for $n = 0, 1, ...10$. Figure 6b illustrates the results for Resnet-20 and Resnet-110, for both a typical (piecewise constant) training regime with a standard initial learning rate of 0.35 and for CLR with a stepsize of 10,000 iterations. For this entire range of architecture depths, super-convergence was possible. The accuracy increase due to super-convergence appears to be greater for the shallower architectures (Resnet-20: CLR 90.4% versus piecewise constant LR schedule 88.6%) than for the deeper architectures (Resnet-110: CLR 92.1% versus piecewise constant LR schedule 91.0%).

Recently, there are discussions in the deep learning literature on the effects of larger batch size and the generalization gap (Keskar et al., 2016; Jastrzębski et al., 2017; Chaudhari & Soatto, 2017; Hoffer et al., 2017). Hence, we investigated the effects total mini-batch size[3] used in super-convergence training and found a small improvement in performance with larger batch sizes, as can be seen in Figure 7a. In addition, Figure 7b shows that the generalization gap (the difference between the training and test accuracies) are approximately equivalent for small and large mini-batch sizes. This result differs than results reported elsewhere (Keskar et al., 2016) and illustrates that a benefit of training with large learning rates is the ability to use large batch sizes (Goyal et al., 2017).

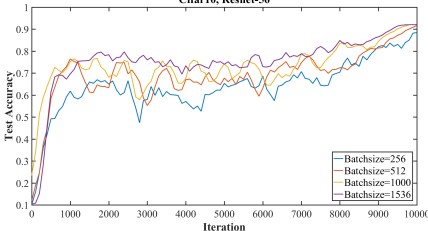

(a) Comparison of test accuracies for Cifar-10, Resnet-56 while varying the total batch sizes.

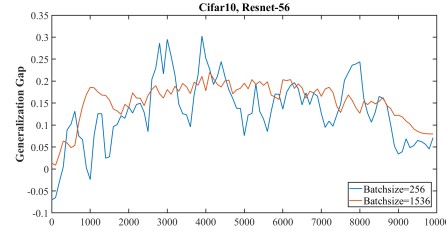

(b) Comparison of the generalization gap (training - test accuracy) for a small and a large batch size.

Figure 7: Comparisons of super-convergence to over a range of batch sizes. These results show that a large batch size is more effective than a small batch size for super-convergence training.

We ran a series of experiments with a variety of ranges for CLR. Table 1 shows the results for maximum learning rate bounds from 1.0 to 3.5. These experiments show that a maximum learning

---

[3]Most of our reported results are with a total mini-batch size of 1,000 and we primarily used 8 GPUs and split the total mini-batch size 8 ways over the GPUs.

rate of approximately 3 performed well. Although our experiments used a minimum learning rate of 0.1, training with a single cycle of CLR and minimum learning rate of 0 performs quite well.

We tested the effect of batch normalization on super-convergence. Initially, we found that having $use\_global\_stats : true$ in the test phase prevents super-convergence. However, we realized this was due to using the default value of $moving\_average\_fraction = 0.999$ that is only appropriate for the typical, long training times. However, when a network is trained very quickly, as with super-convergence, a value of 0.999 does not update the accumulated global statistics quickly enough. Hence, we found that the smaller values listed in Table 1 (column BN MAF) were more appropriate.

We ran a variety of other experiments in which super-convergence continued to occur. We ran experiments with Cifar-100 that demonstrated the super-convergence phenomenon. We found that adding dropout to the architecture still permitted super-convergence and improved the results a small amount. Our experiments also investigated whether adaptive learning rate methods in a piecewise constant training regime would learn to adaptively use a large learning rate to improve performance. We tested Nesterov momentum (Sutskever et al., 2013; Nesterov, 1983), AdaDelta (Duchi et al., 2011), AdaGrad (Zeiler, 2012), and Adam (Kingma & Ba, 2014) on Cifar-10 with the Resnet-56 architecture but none of these methods speed up the training process in a similar fashion to super-convergence. This is important because *it is indicative of the failing in the theory behind these approaches.* We also tried these methods with CLR with appropriate bounds and found super-convergence happens for AdaDelta, AdaGrad and Nesterov momentum but we were unable to achieve super-convergence with Adam. In addition, we investigated the effect of momentum. We ran experiments with momentum in the range of $0.80 - 0.95$ and found momentums between $0.8 - 0.9$ yield a higher final test accuracy. We illustrate the results of these experiments in the supplemental materials.

## 7 DISCUSSION

There is substantial discussion in the literature on stochastic gradient descent (SGD) and understanding why solutions generalize so well (i.e, Chaudhari et al. (2016); Chaudhari & Soatto (2017); Im et al. (2016); Jastrzębski et al. (2017); Smith & Le (2017); Kawaguchi et al. (2017)). Super-convergence provides empirical evidence that supports some theories, contradicts some others, and points to the need for further theoretical understanding. We hope the response to super-convergence is similar to the reaction to the initial report of network memorization (Zhang et al., 2016), which sparked an active discussion within the deep learning research community on better understanding of the factors in SGD leading to solutions that generalize well (i.e., (Arpit et al., 2017)).

Our work impacts the line of research on SGD and the importance of noise for generalization. In this paper we focused on the use of CLR with very large learning rates, which adds noise in the middle part of training. Recently, Jastrzębski et al. (2017) stated that higher levels of noise lead SGD to solutions with better generalization. Specifically, they showed that the ratio of the learning rate to the batch size, along with the variance of the gradients, controlled the width of the local minima found by SGD. Independently, Chaudhari & Soatto (2017) show that SGD performs regularization by causing SGD to be out of equilibrium, which is crucial to obtain good generalization performance, and derive that the ratio of the learning rate to batch size alone controls the entropic regularization term. They also state that data augmentation increases the diversity of SGD's gradients, leading to better generalization. These two papers provide theoretical support for the super-convergence phenomenon. In addition there are several other papers in the literature which state that wide, flat local minima produce solutions that generalize better than sharp minima (Hochreiter & Schmidhuber, 1997; Keskar et al., 2016; Wu et al., 2017). Our super-convergence results align with these results.

In addition, there are several recent papers on the generalization gap between small and large mini-batches and the relationship between gradient noise, learning rate, and batch size. Our results here supplements this other work by illustrating the possibility of time varying high noise levels during training. As mentioned above, Jastrzębski et al. (2017) showed that SGD noise is proportional to the learning rate, the variance of the loss gradients, divided by the batch size. Similarly Smith and Le (Smith & Le, 2017) derived the noise scale as $g \approx \epsilon N/B(1 - m)$, where $g$ is the gradient noise, $\epsilon$ the learning rate, $N$ the number of training samples, and $m$ is the momentum coefficient. Furthermore, Smith et al. (2017) showed an equivalence of increasing batch sizes instead of a decreasing learning rate schedule. Importantly, these authors demonstrated that the noise scale $g$ is relevant to training and not the learning rate or batch size. Keskar et al. (2016) study the generalization gap between

small and large mini-batches, stating that small mini-batch sizes lead to wide, flat minima and large batch sizes lead to sharp minima. They also suggest a batch size warm start for the first few epochs, then using a large batch size, which amounts to training with large gradient noise for a few epochs and then removing it. Our results contradicts this suggestion as we found it preferable to start training with little or no noise and let it increase (i.e., curriculum training), reach a noise peak, and reduce the noise level in the last part of training (i.e., simulated annealing). Goyal et al. (2017) use a very large mini-batch size of up to 8,192 and adjust the learning rate linearly with the batch size. They also suggest a gradual warmup of the learning rate, which is a discretized version of CLR and matches our experience with an increasing learning rate. They make a point relevant to adjusting the batch size; if the network uses batch normalization, different mini-batch sizes leads to different statistics, which must be handled. Hoffer et al. (2017) made a similar point about batch norm and suggested using ghost statistics. Also, Hoffer et al. (2017) show that longer training lengths is a form of regularization that improves generalization. On the other hand, our results show a different form of regularization that comes from training with very large learning rates, which permits much shorter training lengths.

Furthermore, this paper points the way towards new research directions, such as the following three:

1. Characterizing "good noise" that improves the trained network's ability to generalize versus "bad noise" that interferes with finding a good solution (i.e., Zhang et al. (2016)). We find that there is a lack of a unified framework for treating SGD noise/diversity, such as architectural noise (e.g., dropout (Srivastava et al., 2014), dropconnect (Wan et al., 2013)), noise from hyper-parameter settings (e.g., learning rate, mini-batch size), adding gradient noise, adding noise to weights (Fortunato et al., 2017), and input diversity (e.g., data augmentation, noise). Gradient diversity has been shown to lead to flatter local minimum and better generalization. A unified framework should resolve conflicting claims in the literature on the value of each of these, such as for architectural noise (Srivastava et al. (2014) versus Hoffer et al. (2017)). Furthermore, many papers study each of these factors independently and by focusing on the trees, one might miss the forest.

2. Time dependent application of good noise during training: As described above, combining curriculum learning with simulated annealing leads to cyclical application. To the best of our knowledge, this has only been applied sporadically in a few methods such as CLR (Smith, 2017) or cyclical batch sizes (Jastrzębski et al., 2017) but these all fall under a single umbrella of time dependent gradient diversity (i.e., noise). Also, one might learn an optimal noise level while training the network.

3. Discovering new ways to stabilize optimization (i.e., SGD) with large noise levels: Our evidence indicates that normalization (and batch normalization in particular) is the catalyst enabling super-convergence in the face of destabilizing noise from the large learning rates. Normalization methods (batch norm, layer normalization Ba et al. (2016), streaming normalization Liao et al. (2016)) and new techniques (i.e., cyclical gradient clipping) to stabilize training need further investigation to discover better ways to keep SGD stable in the presence of enormous good noise.

Classical physics was insufficient for explaining super-conductivity when it was discovered and pointed to the need for new theories, such as quantum mechanics. Similarly, super-convergence indicates a need for new theories of SGD and generalization.

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

## A    SUPPLEMENTAL MATERIAL

This Section contains the details of the experiments that were carried out in support of this research. It also contains results for several of our more interesting experiments that did not fit in the main text of the paper.

### A.1    DATASETS, ARCHITECTURES, AND HYPER-PARAMETERS

All of the experiments were run with Caffe (downloaded October 16, 2016) using CUDA 7.0 and Nvidia's CuDNN. These experiments were run on a 64 node cluster with 8 Nvidia Titan Black GPUs, 128 GB memory, and dual Intel Xeon E5-2620 v2 CPUs per node and we utilized the multi-gpu implementation of Caffe.

The Resnet-56 architecture used consists of three stages. Within each stage, the same residual block structure is sequentially repeated. This structure is given in Table 2. Between stages, a different residual block structure is used to reduce the spatial dimension of the channels. Table 3 shows this structure. The overall architecture is described in Table 4. Following the Caffe convention, each Batch Norm layer was followed by a scaling layer to achieve true batch normalization behavior. This and the other architectures necessary to replicate this work will be made available upon publication.

Table 2: A residual block which forms the basis of a Residual Network.

| Layer | Parameters |
|---|---|
| Conv Layer 1 | padding = 1
kernel = 3x3
stride = 1
channels = $numChannels$ |
| Batch Norm 1 | moving_average_fraction=0.95 |
| ReLu 1 | — |
| Conv Layer 2 | padding = 1
kernel = 3x3
stride = 1
channels = $numChannels$ |
| Batch Norm 2 | moving_average_fraction=0.95 |
| Sum (BN2 output with original input) | — |
| ReLu 2 | — |

Table 3: A modified residual block which downsamples while doubling the number of channels.

| Layer | Parameters |
|---|---|
| Conv Layer 1 | padding = 1
kernel = 3x3
stride = 2
channels = $numChannels$ |
| Batch Norm 1 | moving_average_fraction=0.95 |
| ReLu 1 | — |
| Conv Layer 2 | padding = 1
kernel = 3x3
stride = 1
channels = $numChannels$ |
| Batch Norm 2 | moving_average_fraction=0.95 |
| Average Pooling (of original input) | padding = 0
kernel = 3x3
stride = 2 |
| Sum (BN2 output with AvgPool output) | — |
| ReLu 2 | — |
| Concatenate (with zeroes) | channels = 2*$numChannels$ |

## A.2   ADDITIONAL RESULTS

We ran a wide variety of experiments and due to space limitations in the main article, we report some of the more interesting results here that did not fit in the main article.

In the main text we only showed the results of super-convergence for the Cifar-10 dataset. In fact, the super-convergence phenomenon also occurs with Cifar-100, which implies an independence of this phenomenon on the number of classes. Figure 8a shows the results of the LR range test for Resnet-56 with the Cifar-100 training data. The curve is smooth and accuracy remains high over the entire range from 0 to 3 indicating a potential for super-convergence. An example of super-convergence with Cifar-100 with Resnet-56 is given in Figure 8b, where there is also a comparison to the results

Table 4: Overall architecture for ResNet-56.

| Layer/Block Type | Parameters |
|---|---|
| Conv Layer | padding = 1
kernel = 3x3
stride = 2
channels = 16 |
| Batch Norm | moving_average_fraction=0.95 |
| ReLU | — |
| ResNet Standard Block x9 | $numChannels = 16$ |
| ResNet Downsample Block | $numChannels = 16$ |
| ResNet Standard Block x8 | $numChannels = 32$ |
| ResNet Downsample Block | $numChannels = 32$ |
| ResNet Standard Block x8 | $numChannels = 64$ |
| Average Pooling | padding = 0
kernel = 8x8
stride = 1 |
| Fully Connected Layer | — |

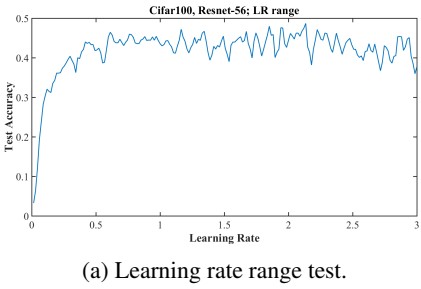

(a) Learning rate range test.

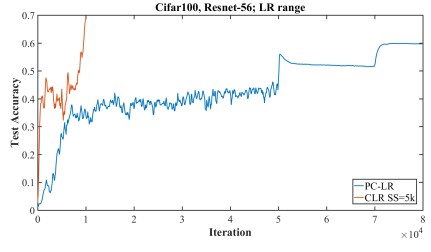

(b) Comparison of test accuracies for super-convergence to a piecewise constant training regime.

Figure 8: Comparisons for Cifar-100, Resnet-56 of super-convergence to typical (piecewise constant) training regime.

from a piecewise constant training regime. Furthermore, the final accuracy for the super-convergence curve is 68.6%, while the accuracy for the piecewise constant method is 59.8%, which is an 8.8% improvement.

As discussed in the main text, we tested various adaptive learning rate methods with Resnet-56 training on Cifar-10 to determine if they are capable of recognizing the need for using very large learning rates. Figure 9a shows the results of this training for Nesterov momentum (Sutskever et al., 2013; Nesterov, 1983), AdaDelta (Duchi et al., 2011), AdaGrad (Zeiler, 2012), and Adam (Kingma

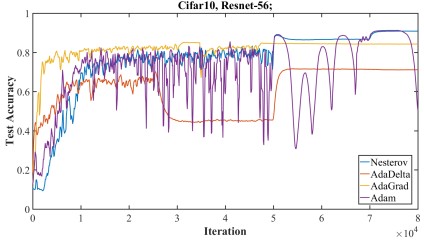

(a) Comparison of test accuracies with various adaptive learning rate methods.

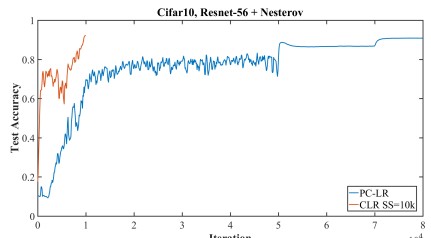

(b) Comparison of test accuracies With Nesterov method.

Figure 9: Comparisons for Cifar-10, Resnet-56 of super-convergence to piecewise constant training regime.

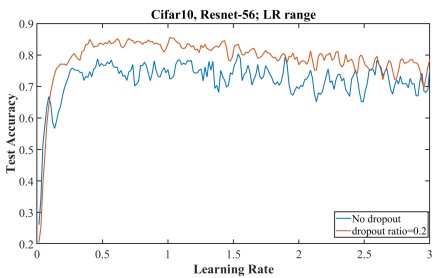

(a) Comparison of LR range test for super-convergence with and without dropout.

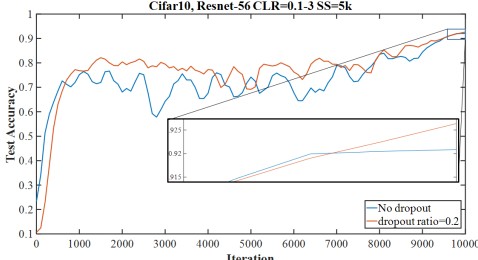

(b) Comparison of test accuracies for super-convergence with and without dropout.

Figure 10: Comparisons of super-convergence with and without dropout (dropout ratio=0.2).

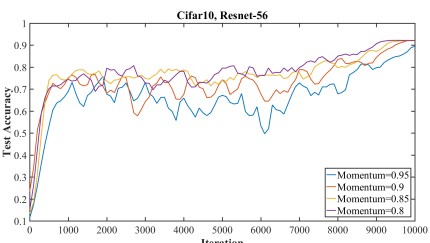

(a) Comparison of test accuracies with various values for momentum.

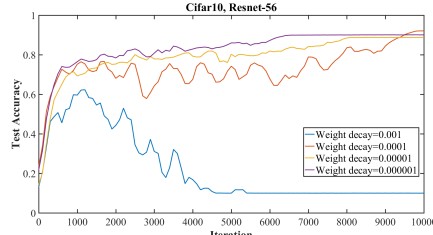

(b) Comparison of test accuracies for super-convergence with various values of weight decay.

Figure 11: Comparisons for Cifar-10, Resnet-56 of super-convergence to typical training regime.

& Ba, 2014). We found no sign that any of these methods discovered the utility of large learning rates nor any indication of super-convergence-like behavior. We also ran CLR with these adaptive learning methods and found that Nesterov, AdaDelta, and AdaGrad allowed super-convergence to occur, but we were unable to create this phenomenon with Adam. For example, Figure 9b shows a comparison of super-convergence to a piecewise constant training regime with the Nesterov momentum method. Here super-convergence yields a final test accuracy after 10,000 iterations of 92.1% while the piecewise constant training regime at iteration 80,000 has an accuracy of 90.9%.

Figure 10 shows a comparison of runs of the super-convergence phenomenon, both with and without dropout. The LR range test with and without dropout is shown in Figure 10a. Figure 10b shows the results of training for 10,000 iterations. In both cases, the dropout ratio was set to 0.2 and the Figure shows a small improvement with dropout. We also ran with other values for the dropout ratio and consistently saw similar improvements.

The effect of mini-batch size is discussed in the main paper but here we present a table containing the final accuracies of super-convergence training with various mini-batch sizes. One can see in Table 5 the final test accuracy results and this table shows that the larger the mini-batch size, the better the final accuracy, which differs from the results shown in the literature[4]. Most of results reported in this paper are with a total mini-batch size of 1,000.

In addition, we ran experiments on Resnet-56 on Cifar-10 with modified values for momentum and weight decay to determine if they might hinder the super-convergence phenomenon. Figure 11a shows the results for momentum set to 0.8, 0.85, 0.9, and 0.95 and the final test accuracies are listed in Table 6. These results indicate only a small change in the results, with a setting of 0.9 being a bit better than the other values. In Figure 11b are the results for weight decay values of $10^{-3}, 10^{-4}, 10^{-5}$, and $10^{-6}$. In this case, a weight decay value of $10^{-3}$ prevents super-convergence, while the smaller values do not. This Figure also shows that a weight decay value of $10^{-4}$ performs well.

---

[4]GPU memory limitations prevented our testing a total batch size greater than 1,530

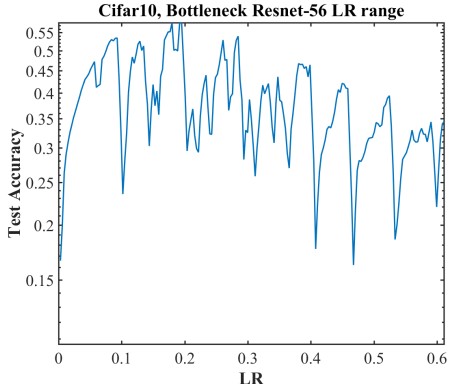
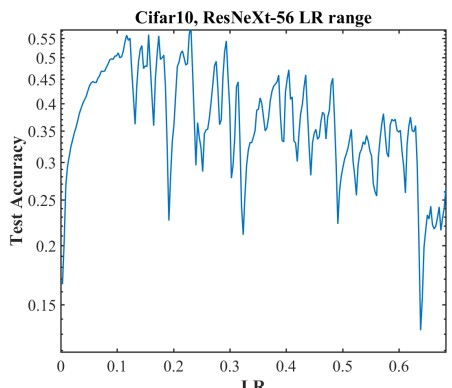

(a) Learning rate range test result with the bottleneck Resnet-56 architecture.

(b) Learning rate range test result with the ResNeXt-56 architecture.

Figure 12: Comparison of learning rate range test results on Cifar-10 with alternate architectures.

| Mini-batch size | Accuracy (%) |
|---|---|
| 1536 | 92.1 |
| 1000 | 92.4 |
| 512 | 91.7 |
| 256 | 89.5 |

Table 5: Comparison of accuracy results for various total training batch sizes with Resnet-56 on Cifar-10 using CLR=0.1-3 and stepsize=5,000.

### A.3 WHERE SUPER-CONVERGENCE DOES NOT OCCUR

We did not observe super-convergence in many experiments, however this does not necessarily mean that the super-convergence phenomenon is impossible in those cases but only that we did not find a way to induce it.

Figure 12a shows an example of the LR range test with a bottleneck version of Resnet-56. There appears to be a peak from a learning rate of approximately 0.15 but even more pronounced are the large swings in the accuracy from around 0.1 and onward. These oscillations and the decay of accuracy at learning rates greater than 0.1 indicate that one cannot train this network with large learning rates and our experiments confirmed this to be true. We also tried Cifar-10 with the ResNeXt architecture and were unable to produce the super-convergence phenomenon. Figure 12b shows an example of the LR range test with ResNeXt. This result is similar to Figure 12a. There appears to be a peak and large swings in the accuracy from a learning rate of approximately 0.1. These oscillations and the decay of accuracy at learning rates greater than 0.1 are indications that one cannot train this network with large learning rates.

Our experiments with Densenets and all our experiments with the Imagenet dataset for a wide variety of architectures failed to produce the super-convergence phenomenon. Here we will list some of the experiments we tried that failed to produce the super-convergence behavior. Super-convergence did

| Momentum | Accuracy (%) |
|---|---|
| 0.80 | 92.1 |
| 0.85 | 91.9 |
| 0.90 | 92.4 |
| 0.95 | 90.7 |

Table 6: Comparison of accuracy results for various momentum values with Resnet-56 on Cifar-10 using CLR=0.1-3 and stepsize=5,000.

not occur when training with the Imagenet dataset; that is, we ran Imagenet experiments with Resnets, ResNeXt, GoogleNet/Inception, VGG, AlexNet, and Densenet without success. Other architectures we tried with Cifar-10 that did not show super-convergence capability included ResNeXt, Densenet, and a bottleneck version of Resnet.

