# OpenReview forum: "Super-Convergence: Very Fast Training of Residual Networks Using Large Learning Rates"
_ICLR.cc/2018/Conference — Reject_

### Official Review · AnonReviewer2 · 2017-11-21
**Interesting Phenomenon, but no deep insights**

**Rating:** 4
**Confidence:** 3

**Review:**

The paper discusses a phenomenon where neural network training in very specific settings can profit much from a schedule including large learning rates. Unfortunately, this paper feels to be hastily written and can only be read when accompanied with several references as key parts (CLR) are not described and thus the work can not be reproduced from the paper.

The main claim of the author hinges of the fact that in some learning problems the surface of the objective function can be very flat near the optimum. In this setting, a typical schedule with a decreasing learning rate would be a bad choice as the change of curvature must be corrected as well. However, this is not a general problem in neural network training and might not be generalizable to other datasets or architectures as the authors acknowledge.

In the end, the actual gain of this paper is only in the form of a hypothesis but there is only very little enlightenment, especially as the only slightly theoretical contribution in section 5 does not predict the observed behavior.

Personally i would not use the term "convergence" in this setting at all as the runs are very short and thus we might not be close to any region of convergence. Most of the plots shown are actually not converged and convergence in test accuracy is not the same as convergence in training loss, which is not shown at all. The results of smaller test error with larger learning rates on small training sets might therefore just be the inability of the optimizer to get closer to the optimum as steps are too long to decrease the expected loss, thus having a similar effect as early stopping.

Pros:
- Many experiments which try to study the effect
Cons:
-The described phenomenon seems to depend strongly on the problem surface and might never
be encountered on any problem aside of Cifar-10
- Only single runs are shown, considering the noise on those the results might not be reproducible.
-Experiments are not described in detail
-Experiment design feels "ad-hoc" and unstructured
-The role and value of the many LR-plots remains unclear to me.

Form:
- The paper does not maker clear how the exact schedules work. The terms are introduced but the paper misses the most basic formulas
- Figures are not properly described, e.g. axes in Figures 3 a) and b)
- Explicit references to code are made which require familiarity with the used framework(if at all published).

---

### Official Review · AnonReviewer1 · 2017-11-27
**Large cyclic learning rates for fast convergence, works in very narrow conditions**

**Rating:** 4
**Confidence:** 4

**Review:**

In this paper, the authors analyze training of residual networks using large cyclic learning rates (CLR). The authors demonstrate (a) fast convergence with cyclic learning rates and (b) evidence of large learning rates acting as regularization which improves performance on test sets – this is called “super-convergence”. However, both these effects are only shown on a specific dataset, architecture, learning algorithm and hyper parameter setting.


Some specific comments by sections:

2. Related Work: This section loosely mentions other related works on SGD, topology of loss function and adaptive learning rates. The authors mention Loshchilov & Hutter in next section but do not compare it to their work. The authors do not discuss a somewhat contradictory claim from NIPS 2017 (as pointed out in the public comment): http://papers.nips.cc/paper/6770-train-longer-generalize-better-closing-the-generalization-gap-in-large-batch-training-of-neural-networks.pdf

3. Super-convergence: This is a well explained section where the authors describe the LR range test and how it can be used to understand potential for super-convergence for any architecture. The authors also provide sufficient intuition for super-convergence. Since CLRs were already proposed by Smith (2015), the originality of this work would be specifically tied to their application to residual units. It would be interesting to see a qualitative analysis on how the residual error is impacting super-convergence.

4. Regularization: While Fig 4 demonstrates the regularization property, the reference to Fig 1a with better test error compared to typical training methods could simply be a result of slower convergence of typical training methods.
5. Optimal LRs: Fig.5b shows results for 1000 iterations whereas the text says 10000 (seems like a typo in scaling the plot). Figs 1 and 5 illustrate only one cycle (one increase and one decrease) of CLR. It would be interesting to see cases where more than one cycle is required and to see what happens when the LR increases the second time.

6. Experiments: This is a strong section where the authors show extensive reproducible experimentation to identify settings under which super-convergence works or does not work. However, the fact that the results only applies to CIFAR-10 dataset and could not be observed for ImageNet or other architectures is disappointing and heavily takes away from the significance of this work.

Overall, the work is presented as a positive result in very specific conditions but it seems more like a negative result. It would be more appealing if the paper is presented as a negative result and strengthened by additional experimentation and theoretical backing.

---

> ### Public Comment · ~Dmytro_Mishkin2 · 2017-12-23
> **Residual is not necessary**
>
> >It would be interesting to see a qualitative analysis on how the residual error is impacting super-convergence.
>
> ResNet is not needed, actually. My experiments with HardNet local descriptor (see me previous public comment)  use plain VGG-like architecture and still achieve some king of "super-convergence".

---

### Official Review · AnonReviewer3 · 2017-12-01
**This paper shows an observation of “super-convergence” when training resnet with cyclical learning rates but does not provide conclusive analysis or experiment results.**

**Rating:** 4
**Confidence:** 3

**Review:**

This paper discusses the phenomenon of a fast convergence rate for training resnet with cyclical learning rates under a few particular setting. It tries to provide an explanation for the phenomenon and a procedure to test when it happens. However, I don't find the paper of high significance or the proposed method solid for publication at ICLR.

The paper is based on the cyclical learning rates proposed by Smith (2015, 2017). I don't understand what is offered beyond the original papers. The "super-convergence" occurs under special settings of hyper-parameters for resnet only and therefore I am concerned if it is of general interest for deep learning models. Also, the authors do not give a conclusive analysis under what condition it may happen.

The explanation of the cause of "super-convergence" from the perspective of  transversing the loss function topology in section 3 is rather illustrative at the best without convincing support of arguments. I feel most content of this paper (section 3, 4, 5) is observational results, and there is lack of solid analysis or discussion behind these observations.

---

### Public Comment · ~Dmytro_Mishkin2 · 2017-10-31
**External validation: both for and against paper results.**

I have done an additional experiement in different domain.

1) Task: local patch descriptor learning.
Architecture: Siamese network, output is 128-channel descriptor, which is L2-normed.
Then triplet margin loss applied to the triplet of : anchor, positive, negative.

2)Networks itself is VGG-style:
32x32 grayscale, locally normalized patch -> 32C3-32C3-64C3/2-64C3-128C3/2-128C8 - L2norm
No residual connections, no bottlenecks, but batch-normalization after each conv layer.

3)Dataset: 5M triplets, randomly sampled from 100K patches from Brown dataset
http://phototour.cs.washington.edu/patches/default.htm

4) lr_rate decay is linear from max_lr to 0 , as it work better than standard "step" one.

5) Metric is mAP two view matching on two other datasets: W1BS and HPatches. So metric really tests generatlization

So, results:

LR policy                                                         | Iterations | mAP

Linear, from 0.1 to 0                                       | 50K          | 0.1065
Linear, from 50 to 0                                        |    5K          | 0.1087
(0.9 * abs(sin)) * + 0.1) *(Linear, from 50 to 0)|    5K          | 0.1100

So I am  not sure, if it can be called "super-convergence" in authors sense, but large learning rate lead to improved performance in my case + "cyclic modulation" makes effect bigger.


First, batch normalization seems necessary part, because it basically allows to have huge weights, which does not influence output. And at the end of my network there is L2norm, so everything is always fine-scaled.

Second, if the large weights are one of the responsible parts, may be recent paper on "Feature Incay" is relevant https://arxiv.org/pdf/1705.10284.pdf
In short, authors argue, that large values of the features contrary to common practice, lead to better generalization. But they don`t tell anything about convergency speed.

Third, unfortunately, result of the faster converged network was _worse_ on real-world with matching.

The last, but not least, this paper contradicts recent NIPS oral "Train longer, generalize better: closing the
generalization gap in large batch training of neural
networks"  https://arxiv.org/pdf/1705.08741.pdf, where authors show that longer training is important for generalization.

Solving this contradiction could lead to new interesting results.


****
Paper overall is good written and opens an interesting discussion. I would vote for poster acceptance

---

### Author Response · Authors · 2017-11-20
**Two new papers (after the submission deadline) independently provide theoretical support for the super-convergence phenomenon**

Jastrz{\k{e}}bski,  et al. [1] show that the larger the ratio of the learning rate to the batch size, the greater the noise during training and the better the network generalizes. They also demonstrate that instead of increasing the learning rate via cyclical learning rates, one obtains a similar effect by decreasing the batch size.  Independently, Chaudhari, et al. [2] show that the entropy of the steady-state distribution of the weights scales linearly with the ratio of the learning rate over two times the batch size and this ratio completely determines the strength of SGD's regularization.  Although the authors don't suggest a cycle, they do recommend that this ratio be large in practice, which coincides with our empirical results.

1. Jastrzębski, Stanisław, Zachary Kenton, Devansh Arpit, Nicolas Ballas, Asja Fischer, Yoshua Bengio, and Amos Storkey. "Three Factors Influencing Minima in SGD." arXiv preprint arXiv:1711.04623 (2017).
2. Chaudhari, Pratik, and Stefano Soatto. "Stochastic gradient descent performs variational inference, converges to limit cycles for deep networks." arXiv preprint arXiv:1710.11029 (2017).

---

### Author Response · Authors · 2017-12-11
**General reply to the Reviewers comments.**

Thank you to all the reviewers for your time and effort in reading our paper.

Although many papers in the deep learning literature suggest new techniques for training deep networks, we did not intend for this paper to be of this kind.  Instead, this super-convergence paper presents empirical evidence of a new phenomenon that is not yet adequately explained by the literature on SGD.  While super-convergence might be of some practical value, the primary purpose of this paper is to provide empirical support and theoretical insights to the active discussions in the literature on SGD and understanding generalization.  Based on the reviewers' comments, it is apparent that the relevance of super-convergence to ongoing discussions in the literature is unclear.  We have rewritten the Discussion Section and revised various other parts of the paper to more explicitly show how our results are relevant to ongoing discussions in the literature on SGD and generalizations.  We hope the response to super-convergence is similar to the reaction to the initial report of network memorization, which sparked an active discussion within the deep learning research community on better ways of understanding the factors in SGD leading to solutions that generalize well.

---

### Author Response · Authors · 2017-12-11
**Replies to specific reviewer’s comments.**

1. AnonReviewer1 comments and replies:
"Loshchilov & Hutter in next section but do not compare it to their work."
We stated that Loshchilov & Hutter's form for CLR (called SGDR) does not work for super-convergence.  The paper now states this more clearly in the Related Works Section.

"contradictory claim from NIPS 2017 (as pointed out in the public comment)"
We do not consider the claims in Hoffer, et al. (2017) contradictory.  They show that a longer training is a form of regularization, which doesn't contradict the regularization effects of large learning rates any more than it contradicts the use of dropout for regularization.  From a practical perspective, training longer has the obstacle of an even larger computational burden, hence other forms of regularization are preferable.

"It would be interesting to see a qualitative analysis on how the residual error is impacting super-convergence."
We don't think it is the residual nature of the networks that are relevant but how batch norm stabilizes the training in the presence of large learning rates causing gradient noise.  We discuss this more clearly now.

"Fig 1a with better test error compared to typical training methods could simply be a result of slower convergence of typical training methods."
Hoffer, et al. (2017) implies that longer training would improve the slower convergence rate in Fig 1a.  We actually let the training for the typical training schedule go to 120,000 iterations but the test accuracy was higher at 80,000 so Fig 1a shows longer training in a better light.

"It would be interesting to see cases where more than one cycle is required and to see what happens when the LR increases the second time."
This has been done.  For example, see "Snapshot ensembles: Train 1, get m for free" arXiv:1704.00109.  Most of our experiments were performed last winter, prior to this paper but we saw similar results as they described.

"Overall, the work is presented as a positive result in very specific conditions but it seems more like a negative result."
The super-convergence paper presents empirical evidence of a new phenomenon that is not yet adequately explained by the literature on SGD, as such it is a positive result.  The Discussion Section should make the impact of this work clearer.

Thank you for your comments and the opportunity to address your concerns.


2. AnonReviewer2 comments and replies:
"the work cannot be reproduced from the paper."
Architectures and code will be available on github.com.

"convergence in training loss, which is not shown at all"
The training loss is shown in Figure 4.  Furthermore, we examined the training loss for all of the figures but did not include them in most of the figures for readability and it did not provide any additional insights.

"-The described phenomenon seems to depend strongly on the problem surface and might never be encountered on any problem aside of Cifar-10"
"- Only single runs are shown, considering the noise on those the results might not be reproducible."
If the purpose of the paper was to demonstrate another new technique to obtain a half a percent improvement in results, we would have averaged over 10 runs to show that the half-percent improvement. Also, the limitation of the effect to only Cifar would heavily detract from the practical significance of this paper. However, that is tangential to the primary purpose of this paper. Instead, this super-convergence paper presents empirical evidence of a new phenomenon that is not yet adequately explained by the literature on SGD and regularization.

"-Experiments are not described in detail."
"-Experiment design feels "ad-hoc" and unstructured"
"-The role and value of the many LR-plots remains unclear to me."
"- The paper does not maker clear how the exact schedules work. The terms are introduced but the paper misses the most basic formulas"
Architectures and code will be available on github.com.

"- Figures are not properly described, e.g. axes in Figures 3 a) and b)"
The caption for Figure 3 was amended.  This figure was borrowed with permission from "Qualitatively characterizing neural network optimization problems." arXiv:1412.6544 (2014) and a full description is available in that paper.

"- Explicit references to code are made which require familiarity with the used framework(if at all published)."
Architectures and code will be available on github.com.

3. AnonReviewer3 comments and replies:
"I don't understand what is offered beyond the original papers."
"I am concerned if it is of general interest for deep learning models."
"Also, the authors do not give a conclusive analysis under what condition it may happen."
"a lack of solid analysis or discussion behind these observations."

We believe the significance of this paper and how it is intertwined with recent discussions in the literature on SGD and generalization is made clearer by the Discussions Section.

---

### Public Comment · ~Josh_Varty1 · 2017-12-14
**Reproduction**

I have chosen to reproduce elements of this paper as part of the ICLR 2018 Reproducibility Challenge: http://www.cs.mcgill.ca/~jpineau/ICLR2018-ReproducibilityChallenge.html

The key claim of this paper that I attempted to reproduce is that a Resnet-56 network can be trained to ~90% accuracy on Cifar-10 in just 10,000 steps with a Cyclical Learning Rate (CLR). I also wanted to confirm the baseline result that it would take 80,000 steps to train the same network to similar accuracy using a traditional multistep learning.

These results were presented in Figure 1A from the paper.

I took two approaches when reproducings this work:

1. I attempted to reproduce the work in Tensorflow using both the paper and the author's Caffe code as a guide.
2. I attempted to reproduce the work using the author's Caffe code and GitHub instructions.


Reproducing with Tensorflow

I have made my code available on GitHub at: https://github.com/JoshVarty/ReproducingSuperconvergence

Using Tensorflow I was able to weakly reproduce evidence of super-convergence.

After 10,000 steps training with CLR, the network achieved ~85% accuracy. See: https://i.imgur.com/e9RXHl1.png
After 20,000 steps training with multistep, the network achieved ~80% accuracy. See: https://i.imgur.com/PGZ9nlI.png

Although these results do not quite align perfectly with those of the paper, I believe they support it. Although multistep training was run for 80,000 steps it did not  improve after the first 20,000 steps. I was also unable to achieve accuracies over 90% as shown in the paper. I believe this may be due to the fact I was only able to use a mini-batch size of 125 compared to the author's mini-batch size of 1,000.


Reproducing with Caffe

Using the provided Caffe code, I was able to partially reproduce the results presented in the paper.

For baseline multistep learning, I achieved a test accuracy of 85%. See: https://i.imgur.com/8SaqJJ3.png
For CLR learning, I achieved a test accuracy of 91.2%. See: https://i.imgur.com/zVds4VF.png

The overall trend looks similar to that of the author's results, but the test accuracy of CLR does not quite match the expected results presented in the paper.

Potential reasons for lack of reproduction
	- The author trained their network using an 8-GPU machine with a mini-batch size of 1,000. I used a batch size of 125 on a single K80 GPU.
	- Difference in Batch Normalization parameter settings. Currently investigating this here: https://github.com/lnsmith54/super-convergence/issues/2


Corrections

	- Appendix A claims the first Conv Layer has stride=2, but the code provided uses stride=1.


Undocumented Elements

Some elements of the network were undocumented in the paper making it harder to reproduce:

	- While training, images were flipped left-to-right with 50% probability
	- All weights before ReLUs are initialized according to "Delving Deep into Rectifiers." [1]
	- All weights before softmax are initialized according to "Understanding the difficulty of training deep feedforward neural networks." [2]
	- Bias variables are initialized to zero.
	- Learning rate scaling on weights layers was 1
	- Learning rate scaling on bias layers was 2


Conclusion

There is evidence to suggest that super-convergence reproduces in some form on Cifar-10 with a Resnet-56 architecture. On a personal note, I will be incorporating Cyclical Learning Rates into future projects of mine.


[1] https://arxiv.org/pdf/1502.01852v1.pdf
[2] http://citeseerx.ist.psu.edu/viewdoc/download?doi=10.1.1.207.2059&rep=rep1&type=pdf

---

### Decision · Program_Chairs · 2018-01-29
**ICLR 2018 Conference Acceptance Decision**

**Decision:**

Reject

**Comment:**

The paper reports unusally rapid convergence of the ResNet-56 model on CIFAR-10 when a single cycle of a cyclic learning rate schedule is used.  The effect is analyzed from several different perspectives. However, the reviewers were not convinced because the effect is only observed for one task, so they question the significance of the result. There was significant discussion of the paper by the reviewers and area chair before this decision was reached.

Pros:
+ Paper illustrates a "super-convergence" phenomenon in which training of a ResNet-56 reaches an accuracy of 92.4% on CIFAR-10 in 10,000 iterations using a single cycle of a cyclic learning rate schedule, while a more standard piecewise-constant schedule reaches 91.2% accuracy in 80,000 iterations.
+ There was partial, independent replication of the results on other tasks reported on OpenReview.

Cons:
- In the paper, the effect is shown for only one architecture and one task.
- In the paper, the effect is shown for only a single run.
- There are no error bars to indicate which differences are significant.